# Street Children in Iran: What Are Their Living and Working Conditions? Findings from a Survey in Six Major Cities

**DOI:** 10.3390/ijerph20075271

**Published:** 2023-03-27

**Authors:** Meroe Vameghi, Payam Roshanfekr, Gholamreza Ghaedamini Harouni, Marzieh Takaffoli, Giti Bahrami

**Affiliations:** 1Social Welfare Management Research Center, University of Social Welfare and Rehabilitation Sciences, Tehran 1985713871, Iran; 2Social Determinants of Health Research Center, Alborz University of Medical Sciences, Karaj 3149969415, Iran

**Keywords:** street children, socio-demographic characteristics, working situation, street challenges, child labor

## Abstract

Street children are among the most marginalized children, globally, who experience severe violations of their rights and face multiple deprivations. This study aimed to describe street children’s characteristics and working conditions in Iran. Method: This cross-sectional rapid survey was conducted from March to May 2017 in six major cities in Iran. The sample group consisted of Iranian and non-Iranian girls and boys, aged 10 to 18, who worked on the streets for at least one month prior to the survey. Time–location based sampling was used. A total of 856 Children were randomly selected from 464 venues, including corners of streets, parks, metro gates, bus stations, shopping malls, and shopping centers frequented by street children. Results: Findings showed that 90% of participants were boys, 60% were between 10 and 14 years old, almost 50% attended school, 12% were illiterate, and 32% had quit school. Children of Afghan nationality comprised 54% of the study participants, and the rest were Iranian. Of all participants, 85% resided with family or relatives. Most children (75.5%) worked more than 5 h daily, and vending (71.2%) and waste picking (16.1%) were common activities. Street children suffered, mainly, from harsh weather (22.7%), insults and beatings of everyday people (21%), starvation (20.7%), and police repression (15.4%). More than half of the study participants were not involved in intervention programs, and just 7% of them had attended any health education programs. Conclusion: Street children reported little to no service use, which may contribute to poor health. Street children require immediate attention to improve their wellbeing. Decision-makers and academicians should collaborate on intervention development research to design appropriate health and social interventions targeted at street children.

## 1. Introduction

The emergence and increase in street children worldwide, especially in developing countries, attest to social, economic, and political problems in societies. The United Nations defines street children as “boys and girls for whom the street (including unoccupied dwellings, wasteland, etc.) has become their home and/or source of livelihood and who are inadequately protected or supervised by responsible adults.” The definition suggests that a child in the street may be a working child, a school dropout, or a homeless boy or girl [1].

In 2005, UNICEF declared that it is not possible to enumerate accurate numbers of street children, but it is most likely that there has been an increase in parallel to population growth, internal migration, and urbanization [2]. Subsequent studies, which have attempted to quantify the phenomenon, have revealed that the number of children working on the streets is increasing worldwide, especially in developing countries [3,4,5].

Children work and/or live on the streets for various reasons, including family factors, socioeconomic situations, and peer effects. Furthermore, these reasons can vary depending on whether children are in industrial or developing countries [6]. Specifically, many global and local studies in countries, including India, Bangladesh, Namibia, Botswana, Cameroon, Democratic Republic of Congo, Egypt, Ethiopia, Ghana, Kenya, Nigeria, Rwanda, Senegal, South Africa, Sudan, Tanzania, Tunisia, Uganda, and Zimbabwe, have identified poverty as the most common reason [7,8,9,10,11,12,13], along with family problems and breakdown, parental deaths, social and cultural norms, displacement due to natural disasters and conflict, domestic violence, child neglect and abuse, inadequate education, legal systems, and peer pressure [5,8,10,11,12,14,15,16,17].

Moreover, street children, globally, are amongst the most marginalized children who experience severe violations of their rights and face multiple deprivations [18] that could affect their health and well-being. One of the main discriminations is the lack of access to essential services and basic needs such as health care and education. Studies in different countries, including Indonesia, Bangladesh, Nepal, and several African countries, reveal that discrimination subjects street children to hunger, malnutrition, body pains, sickness, and exhaustion; furthermore, they are frequently exposed to injuries, burns, accidents, and polluted environments, and they also experience lower academic achievement [10,11,19,20,21,22,23,24,25]. In terms of mental health, evidence from Ethiopia, Ghana, Kenya, Vietnam, South Africa, and India shows that street children experience poor mental health and face challenges such as high levels of stress, anti-social behavior, anxiety, aggression, and depression [6,11,21,26,27,28], and they consume drugs and alcohol [11,29,30,31]. Furthermore, they experience sexual abuse, violence, and risk behaviors that could increase the risk of sexually transmitted infections [5,6,11,32].

Various interventions and practices are implemented in different countries to support street children; notably, these include promoting educational interventions and policies to prevent children from living and working on the streets [33,34,35,36,37]. Generally, five action areas focused on three levels of action—global, local, and people—are recommended by the United Nations (UN) to promote long-term gains for street children: 1—poverty reduction, enhancing social protection, and prevention of family separation; 2—strengthening child protection policies, programs, and services; 3—improving educational outcomes of children and skills development by equal access to justice, guaranteeing quality, and inclusive education; 4—making cities safer for children; 5—investments in evidence-generation [18]. Finally, the UN emphasizes that practices should change from a charitable to a rights-based approach, working with children rather than for them, thus promoting children’s participation in policies and programs that affect them [38].

The Islamic Republic of Iran is an upper-middle-income country in western Asia with a population of more than 80 million, of which 24,151,000 are children (28.8%) [39]. While Iran has anecdotally reported increasing numbers of street children in recent decades, there are no accurate reports on this population at the national level. Regarding the last population estimate of these children, in 2017, the total number of street children in Iran was 26,000 (IQR 20,239–34,719) [40].

The demographics of street children in Iran have been explored in previous studies. Some studies in different cities and provinces indicated that more than 90% of the street children were boys [41,42]; on the other hand, others reported that the share of girls was more than 25% [43,44]. Besides, a study in four provinces showed that 35.1% of street children were 6–11, and 32.25% were 12–14 years old [42]. Considering the share of migrants and refugees, based on a review of the socioeconomic status of street children in Iran, most of the children were Afghan, followed by Iraqis and Pakistanis [45]. Moreover, a study in Tehran, the capital of Iran, revealed that non-Iranians comprised 36.3% of all street children in 2012 [44]. However, a study in four provinces of Iran reported that non-Iranians comprised 16.1% in 2015 [42], while another reported a sample of 100% Iranian children in a 2017 study in Kermanshah city [41]. In terms of family status, living with unemployed, illiterate, and drug user fathers was prevalent among street children [41,44].

Furthermore, following international concerns and evidence, recent studies in Iran indicated that street children are vulnerable to multiple adverse health and social welfare outcomes, including physical violence, sexual violence, drug use, smoking, lack of education, and extreme poverty [43,46,47,48,49]. There was one study that revealed Iranian street children have a significantly higher prevalence of alcohol and drug use than Afghan street children, while Afghan street children have lower access to education and health services [45].

In national laws, such as the Constitution of the Islamic Republic of Iran, Labor Law, 2020 Children and Adolescents Protection Act, and the Convention on the Rights of the Child, there are some specific interventions related to street children since 1999, including establishing street children centers (compulsory boarding centers, non-compulsory child care centers or shelters, and supportive daycare and family-based centers, respectively), developing protocols to gather, managing and supporting them, development of regulation to organize street children by Iran’s Council of Ministers (2004), developing the participation and collaboration of NGOs, and establishing schools and educational centers for working and street children [47]. However, these policies have been criticized because these did not have a significant outcome on their education and secondary needs [42,50,51].

It should be noted that street children are a hidden vulnerable population and a global public health issue, but little is known about their working and living conditions and the mechanisms contributing to their poor health [28,52]. In Iran, most research about children and adolescents has focused on school, community, or clinical-based samples and rarely on vulnerable street children. On the other hand, it is now being observed that the number of street children is significantly increasing; however, there is scattered and contradictory information about street children in various parts of Iran, and little is known about the prevalence of the problem, including the factors that lead to being a street child and their health status [11]. In fact, despite measures and efforts to manage the issue and mitigate the effects of living and working on the streets among children, there still needs to be clear and accurate information about street children at the national level in Iran.

Hence, with a lack of research to guide intervention or program and policy planning, this study aimed to address these research gaps in the literature, add new evidence, and understand street children’s characteristics and working conditions in Iran. This study aimed to describe street children’s characteristics, the reasons children end up on the street, their working conditions and occupational exposures, experiences of abuse, and use of services in six major cities in Iran. Findings can be used to inform interventions targeted at street children.

## 2. Method

### 2.1. Sampling and Design

This cross-sectional survey was conducted from March to May 2017 in six major cities of Iran, including Tehran, Mashhad, Karaj, Kermanshah, Bandar Abbas, and Zahedan, to measure the socio-demographic characteristics, work characteristics, and problems of street working children in Iran by using a rapid assessment and response method. The study population consisted of Iranian and non-Iranian children who spent considerable time of the day in the streets to work and/or live. The study participants were recruited through Time-Location Sampling (TLS) [53], also called time-space sampling or venue-based sampling based on place and time. Place refers to spaces where the target population is present, and times refer to specific days and periods when the target population gathers in each place. These places and days are classified into standard space-time sections (4 h intervals per space) and are known as venue-daytime or VDT units. The purpose of the qualitative phase was to create a list of all potential venues where street children could be found and identify the days and time periods when the maximum number of street children are present. Key informants included diverse persons with knowledge of street children at the city level (e.g., municipal social welfare personnel, public and non-governmental service providers, academicians), as well as the street or venue level (e.g., street children themselves). In practice, the group discussions with street children corroborated venues named by other key informants and added venues not previously identified in each city. Venues solicited included corners of streets, parks, metro gates, bus stations, shopping malls, or centers frequented by street children. 

Key informants and group discussants identified 370 venues (ranging from 23 to 113 per city) where street children were purportedly present. Interviews with street children in the field identified an additional 94 venues (ranging from 4 to 32 across cities). Of the 464 total venues, our team visited 226 (48.7%) at the randomly selected venue-day-times for the 1 h counting periods, as well as 200 venues (43.1%) for the 4 h periods [40].

Based on eligibility criteria, the participants included 10 to 18-year-old children who worked (any work) or had lived for a few hours per day, for at least a month, in the streets. We selected a random sample of 464 venue-daytimes from the available venues, including corners of streets, parks, metro gates, bus stations, shopping malls, and shopping centers frequented by street children. This sample list was created during the qualitative phase of the research, and venues were visited to ensure the presence of children and safety issues. The qualitative phase of the study is detailed elsewhere in Vameghi, Roshanfekr [40].

Data were gathered using a structured questionnaire based on the prior instrument used in a previous study by the authors [54]. The questionnaire included modules on socio-demographic characteristics, characteristics of work, problems of working on the street, and responses or intervention programs provided by governmental and non-governmental institutions. A total of 856 street children were interviewed. The response rate was 95.1%.

### 2.2. Variables 

The current descriptive analysis explores the status of living and working situations among street children, defined as boys and girls aged under 18 years, for whom “the street” (including unoccupied dwellings and wasteland) has become home and/or their source of livelihood and who are inadequately protected or supervised [55]. The participants included 10 to 18-year-old children who worked (any work) or had lived in the streets for a few hours per day for at least a month.

There were six health concerns of interest related to occupational exposures, injuries, and experiences of abuse that were treated as binary variables, including (1) accidents while working on the street, (2) being insulted by everyday people, (3) police, or other agencies, (4) experiencing extreme heat and cold temperatures, (5) sexual abuse, and (6) starvation. The recall period for these questions was 12 months.

Background characteristics and living conditions included gender, age, nationality, marital status, school enrolment, educational status, stage of school dropout, children’s identity document, sleeping place in recent months, the living condition of parents, fathers’ job status, mothers’ job status, fathers’ educational status, fathers’ drug use, and mother’s drug use, and terms of the working condition include working hours (minutes), age of entering the street (years), kind of work, whether a child has an employer, working in places other than streets, arrests in the past three months, and type of street jobs. 

### 2.3. Analysis 

Descriptive analysis (frequencies and percentages) was conducted. Descriptive analysis (frequencies and percentages) was conducted using SPSS-22 software.

### 2.4. Ethics

Participants were informed about the purpose of the survey, the voluntary nature of their participation, incentives, and the anonymity of all collected data. Trained interviewers administered the questionnaires. Interviews were performed by their verbal consent (marked in the questionnaire). Street children attending group discussions received food and refreshments. Street children interviewed at venues received 50,000 Rials (USD 1.35 in 2017) for their time. The study protocol and procedures were reviewed and approved by the Ethics Committee (code: TR.USWR.REC.1395.373) and the Research Review Board at the University of Social Welfare and Rehabilitation Sciences (USWRS).

## 3. Results

### 3.1. Socio-Demographic Characteristics and Living Condition 

Of the 856 study participants, 90% were boys, 60.9% were 10–14-year-old, and the remaining 39.1% were aged 15–18. The proportion of Iranian and Afghan children was 46.1% and 53.9% respectively, and 75.7% of all children had identity documents. Of the study participants, 48% were out of school, and of this group, 24.8% had never attended school (Table 1). Most out of school children (94.7%) quit education in the elementary and middle school periods. The most common reasons children reported for not attending school included the necessity to work (80%), family inability to afford their education costs (32.2%), uninterested in school (18.4%), and prohibition of their family from going to school (10.8%). Regarding the percentage of children enrolled, 52.9% of boys and 46.4% of girls were enrolled in school.

Almost 76% of children’s parents lived together, and the divorce rate among them was 9%. Considering the place to sleep, about 85% of the children were home with their family and relatives a month before the study (Table 1). Street children were from families with low socio-economic status, as 44.7% of fathers and 59.1% of mothers were illiterate, and the unemployment rate among fathers was 29%. Meanwhile, the experience of drug use among fathers was 27.5%, of which 16.6% were current drug users (Table 1). Some children (14.6%) reported leaving home, prompted by different types of abuse (70%), household low economic status (42.5%), parental addiction (15.7%), and parental divorce (14.1%).

### 3.2. Working Conditions

As the findings show, many street children experienced more than one type of street work. Vending (71.2%) and waste picking (16.1%) were the most common occupations. Meanwhile, activities such as drug and alcohol dealing (1.5%) and prostitution (1.1%) are not common among street children (Table 2). Most of the children (52.4%) started working in the street when they were between 10 and 14 years old, and the age of entering the street for 4.2% of children was under 5 years old. Most children (75.5%) worked more than 5 h per day.

The most common reasons that forced children onto the streets were earning money for their families (85.8%) and personal needs (26.9%). Furthermore, making their education cost (10.4%) and families being forced to work (8.9%) were other important factors (Table 2). In addition, in the six months before the study, some street children (13.5%) had experienced working in places other than streets, such as shops or restaurants. According to findings, 70.6% of children did not have an employer, and almost 85% gave all or a part of their income to their families (Table 2).

Moreover, arrest by government authorities is a common experience for children; nearly 45% of them at least had an experience of being arrested by the municipality, police department, or State Welfare Organization (SWO) when working in the streets (Table 2).

### 3.3. Occupational Exosures, Injuries and Experiences of Abuse Faced by Street Children

Children suffered from hot and cold temperatures (22.7%), starvation (20.7%), insults and beatings of everyday people (21%), and police repression (15.4%) when they were working on the streets. Besides, 13.8% of the study participants experienced a car accident while working on the street. In addition, about 1.6% of participants experienced sexual abuse, and 3% did not have a place to sleep (Table 3).

### 3.4. Interventions and Services

In the last year, most children (67.8%) said they received no support from governmental and non-governmental childcare centers and services. Among different services, the most common were State Welfare Organization centers, including Social Emergency Centers and street children centers. On the other hand, only 1.4% of them used NGOs’ facilities and support. Regarding the type of services, more than half of the study participants did not use any services, 24% experienced education, 9% used medical and dental services, and 7% attended health education programs (Table 4).

## 4. Discussion

This study explored the living and working conditions of children in six major cities in Iran. The study’s findings showed that street children generally belong to low-educated and low-income families working in the streets to help their families survive. As a result of living in a poor socioeconomic family situation, entering into work on the streets at a young age, and long hours of working in the streets, mainly without the supervision of parents or other responsible adults, many of them lost the chance of schooling (or attending school along with laborious work), depriving children of play and rest, while they confront verbal and physical violence by people and governmental agents, poor nutrition, and accidents, all of which affect physical and/or mental health. Street children in Iran may fall under the ILO (The International Labour Organization) definition of child labor, as working on the streets in some situations deprives them (any person under 18) of their childhood, potential, and dignity, which harms their physical and mental development. The child labor definition also refers to work that is mentally or morally dangerous, harmful to children, and/or interferes with their schooling by depriving them of the opportunity to attend school, obliging them to leave school prematurely, and requiring them to attempt to combine school attendance with excessively long and heavy work [56]. In our study, many street children quit school at young ages because of the need to work and the lack of funds to pay for education. These findings are consistent with a systematic review of street children’s studies in Iran [45] and many studies in other countries [57,58,59,60]. Moreover, most of the children in our study are boys, although the share of girls decreases with age, which is consistent with other studies in Iran [61,62]

Considering the nationality of street children, there have been some claims from authorities of the government that all or most street children in Iran belong to Afghan immigrant families, which the government has less responsibility for, especially those residing in Iran illegally. However, this study and other studies show that Iranian children make up a considerable share of street children [41,42]. Another dominant discourse in Iran’s media and authorities’ interviews is that illegal gangs and employers systematically exploit street children; meanwhile, findings of this study and others in Iran revealed that most street children have families and are living with them [41,42,63]. Consequently, policies and programs should focus on supporting families and avoiding interventions such as forcefully gathering street children and keeping them in centers based on false beliefs (e.g., children working for organized gangs).

As our findings show, street children in Iran struggle with many health outcomes and challenges on the streets, such as abuse and maltreatment by people and unintentional injuries such as car accidents, as acknowledged by studies from other countries [11,32,38,64]. In line with the findings, and based on a systematic review of studies from low and middle-income countries, street children reported widely ranging rates of survival sex, which is the practice of exchanging sex for money, drugs, shelter, or protection [32]. Street children in Iran also have challenging and abusive experiences of being arrested by different organizations, such as municipalities and State Welfare Organization, to gather and support them. This fact is acknowledged in studies of various countries [32]. However, being arrested causes revictimization of these children and should be considered seriously.

Comparing the health-related outcomes with the general population of children in Iran could give us a clearer view of the risks and challenges that these children experience. While starvation was experienced by more than 20% of street children in this study, there is not the same subjective indicator at the national level to compare with the general population of children in Iran. However, the national malnutrition indicators could be reviewed to elaborate on the findings. Malnutrition refers to a wide range of disorders, including being underweight, wasting, and stunting on the one hand and being overweight and obesity on the other [65]. Malnutrition is not the same thing as starving, although they often go together. According to 2020 data from 13 million students, 6% suffered from wasting (a form of malnutrition) and extreme wasting [66,67]. Finally, the 2017 survey showed that the prevalence of underweight among children under 5 years old was 4.3% [68]. Due to our result, prevalence of experienced starvation is more than malnutrition indicators among children. In this study a fifth of street children reported being starving. Our findings suggest that malnutrition among street children should be further researched.

About 14% of street children experienced car accidents in this study. According to the latest available data from IrMIDHS, a nationwide survey on Iranian rural and urban households in 2010, approximately 0.9% of children (under 18 years) had been involved in at least one road traffic crash resulting in injury [69]. Road accidents are the primary cause of death due to unintentional incidents among Iranian children. According to studies conducted between 2005 and 2018, approximately 60% of fatalities caused by incidents in the 0–19 age group have been due to road accidents [70].

Lastly, regarding the experiences of violence among street children (by people in the streets or authorities), there is no available accurate official statistic at the national level in Iran for child maltreatment. The only available unofficial national data indicate that domestic violence, including child abuse, disability abuse, elder abuse, and parental abuse, accounted for about 30% of the admissions to the social emergency of the State Welfare Organization in 2017. In addition, 40% of domestic abuse cases were related to child abuse, half of which (50%) were due to neglect, and 30%, 16%, and 4% were related to physical, psychological and emotional, and sexual abuse, respectively. It is noteworthy that fathers, mothers, and strangers were the perpetrators in 57%, 26%, and 1.5% of cases, respectively [71]. Considering these statistics, we could somehow compare the prevalence of child abuse by strangers to the experiences of street children in the streets. Moreover, in this study, street children experienced violence from police and other authorities, which is also noted by UNICEF [72]; this group of children is in danger of facing discrimination, violence, and even revictimization in contact with the law if the legal procedure is not child-sensitive.

Besides, as findings show, street children would benefit from educational, supportive, medical, and other services that address their health and empower the children and their families. Reviewing the available services and interventions for these children reveals that, although they mainly focus on primary welfare services to address the basic needs of street children, more than 20% of children experienced issues related to food and clothing, and even a small percentage (3%) have no place to sleep, which could face them to threatening situations. In terms of violence and child protection issues, especially sexual abuse, preventive and self-care education and counseling are among the services provided for these children, mainly through NGOs. However, there is a serious lack of professional trauma and psychological services for maltreated children, a lack of training and advocacy for the rights of these children to develop the child-friendly approach in police and other authorities, and a lack of public awareness about the rights of these children and how to treat them. Furthermore, the lack of access to supportive centers, lack of available services, challenges in providing professional and person-in-environment services, and low-quality of services are mentioned in other studies [42,73]. These findings emphasize the necessity of policies and programs that approach low-income Iranian and Afghan families. Policies and programs should focus on child labor prevention and reinforce governmental and non-governmental harm reduction, as well as family welfare policies and projects.

In this study, we were able to draw a picture of the situation of street children in Iran. Nonetheless, some study limitations were also observed. Firstly, the study’s data collection was self-reported, which is prone to underreporting (recall bias). Secondly, we used Time Location Sampling, which is non-probability sampling. Thirdly, given the hidden nature of child labor, some children might not fully disclose the nature of relationships and the extent of their engagement in labor activities. A low proportion (1.6%) of street children reported sexual abuse. However, studies elsewhere show that disclosure is determined by a complex interplay of factors related to child characteristics, family environment, community influences, and cultural and societal attitudes, and sexual abuse is typically underreported [74]. Our study has other limitations. In addition to missing working and temporarily institutionalized children, we may also underestimate children who spend more time indoors (e.g., clubs and squats). Due to safety issues, we did not visit some venues in Zahedan and Mashhad. Moreover, we only studied six major urban areas in Iran. Therefore, generalization to other urban areas and to rural settings in Iran is limited and results should be interpreted with caution. Nonetheless, this study presents new findings from a large survey of street children in Iran, offering insights into their living and working conditions and use of services, which can inform outreach and interventions with this group.

## 5. Conclusions

Contrary to dominant labels and political discourse, we found that street children in Iran are mainly concerned with supporting their families. Therefore, supporting low-income families through interventions in different levels of policy-making, as well as planning and improving child education, which could prevent pushing children into the streets for more income, seems necessary. Revising policies and approaches regarding immigrants’ rights and their access to jobs and education should also be considered. Unfortunately, the poor economic situation of Iran in recent years has intensified the crisis for low-income families. Findings point to designing and implementing interventions to provide less harmful and protective working environments, monitoring street children’s health, and monitoring their access to health facilities and health education. Enhancing general public awareness regarding street children and professional training of responsible agents, including the State Welfare Organization, police, and NGOs, should be considered. Further studies focused on interventions for specific problems, such as violence, and evaluating them in coordination with local partners and research bodies would also be a step forward.

## Figures and Tables

**Table 1 ijerph-20-05271-t001:** Socio-demographic and living condition characteristics of street children and their parents in six provinces of Iran (*n* = 856).

Variables	Category	Frequency	Percentage
Age (*n* = 865)	10 to14 years old	521	60.9
	15 to 18 years old	335	39.1
Sex (*n* = 865)	Male	767	89.6
	Female	84	9.8
	No answer	5	0.6
Nationality (*n* = 856)	Iranian	395	46.1
	Afghan	461	53.9
Marital status	Never married	826	96.5
	Married (cohabitation and temporary married)	25	2.9
	No answer	5	0.6
School enrolment (*n* = 865)	Yes	431	50.4
	No	411	48
	No answer	14	1.6
Out-of-school children (*n* = 411)	Never attain school	102	24.8
	Dropout	285	67
	No answer	24	5.8
Stage of school dropout (*n* = 285)	Elementary school	132	46.3
	Middle school	138	48.4
	High school	12	4.2
	No answer	3	1.1
Fathers’ job status	Employed/retired	502	58.6
	Unemployed	248	29
	Passed away	85	9.9
	Don’t know	21	2.5
Mothers’ job status	Employment/ retired	116	13.6
	Household	699	81.7
	Others	37	4.3
	Don’t know	4	0.5
Fathers’ educational status	Illiterate	383	44.7
	Elementary	318	37.1
	Secondary grades and diploma	98	11.4
	Upper diploma	6	0.7
	Religious school	3	0.4
	No answer	48	5.6
Mothers’ educational status	Illiterate	506	59.1
	Primary grades	259	30.3
	Secondary grades and diploma	56	6.5
	Upper diploma	5	0.6
	Religious school	1	1
	No answer	29	3.4
The living conditions of parents (*n* = 856)	Married and live together	647	75.6
	Divorced and leaved family	74	8.6
	Death of one or both parents	118	13.8
	No answer	17	2
Father’s drug use	Yes	142	16.6
	No	520	60.7
	Quit	93	10.9
	Don’t know	50	5.8
	No answer	51	6
Mother’s drug use	Yes	29	3.4
	No	747	87.3
	Quit	15	1.8
	Don’t know	17	2
	No answer	48	5.7
Sleeping place in past month	Family and relatives’ home	726	84.8
	Friends’ home	41	4.8
	Shelters	8	0.9
	Workplace, parks, and streets	30	3.6
	More than one place	51	6
Children’s Identity document	Yes	648	75.7
	No	186	21.7
	Don’t know	22	2.6

**Table 2 ijerph-20-05271-t002:** Working conditions of street children in six provinces of Iran (*n* = 856).

Variables	Category	Frequency	Percent
Type of street jobs in the month before the study (multiple responses permitted)	Vending	603	71.2
	Porter	72	8.5
	Busboy	45	5.3
	Weigh people on scales with payment (It is a kind of street job in which somebody provides a weighting scale and sits on the walkway and weighs people and earns a little money).	27	3.2
	Shoe Polishing	27	3.2
	Waste picking	136	16.1
	Dried bread picking	41	4.8
	Begging	27	3.2
	Cleaning car windscreens	17	2
	Pick pocketing	4	0.5
	Sex-worker	9	1.1
	Drug/alcohol distribution	13	1.5
	Vagrancy (without regular employment)	7	0.8
	Other	18	2.1
Most important reasons for working (multiple responses permitted)	Family support	729	85.8
	Personal needs	229	26.9
	Education cost	88	10.4
	Forced by family to work	76	8.9
	violence in the family	34	4
	Not interested in education	13	1.5
Has an employer	Yes	220	25.7
	No	604	70.6
	No answer	32	3.7
Hours worked in street per day	Up to 2 h	13	1.5
	2–5 h	173	20.2
	More than 5 h	646	75.5
	No answer	24	2.8
Who spends daily income earned by child	Family	441	51.5
	Family and child	282	32.9
	Employer and child	107	12.5
	Other	17	2
	No answer	9	1.1
Working in places other than streets	Yes	116	13.6
	No	704	82.2
	No answer	36	4.2
Agency that has arrested child in past three months (multiple responses permitted)	Municipality	266	31
	Police department	127	14.8
	State Welfare Organization	135	24.3
	Not arrested	459	53.6
	No answer	17	2

**Table 3 ijerph-20-05271-t003:** Occupational exposure, injuries, and experiences of abuse faced by street children in six provinces of Iran (*n* = 856).

Variables	Category	Frequency	Percent
Occupational exposure, injuries and experiences of abuse (multiple responses permitted)	Hot and cold temperatures	194	22.7
	Insults and beatings by everyday people	180	21
	Starvation	177	20.7
	Insults and beatings of police and other organization agents	132	15.4
	Car accident	113	13.8
	Sexual abuse	14	1.6
	No place to sleep	26	3

**Table 4 ijerph-20-05271-t004:** Child centers and intervention programs relevant for street children in six provinces of Iran (*n* = 856).

Variables	Category	Frequency	Percentage
Support from children’s centers (last year) (multiple responses permitted)	Social emergency centers of State Welfare Organization	96	11.2
	Street children centers of SWO	108	12.6
	NGOs centers for street children	12	1.4
	Transit Centers (DIC) (Transition Center (DIC): To control or reduce high-risk behaviors, the center provides harm reduction services (healthy injection and healthy sexual behavior training, needle delivery) to people with substance abuse problems and hard-to-reach addicts who do not visit treatment centers).	43	5
	Social services centers	37	4.3
	Correction and Rehabilitation Centers	16	1.9
	The police force (police110 (Police 110 is a dispatcher center of Law Enforcement Command of Islamic Republic of Iran which sends emergency responder units))	39	4.6
	Other	38	4.4
	None	581	67.8
Services used (other) (multiple responses permitted)	Educational programs	205	24
	Life skills education	45	5.3
	Health education	60	7
	Daily food programs	29	3.4
	Technical & Vocational Training	26	3
	Medical and dental Examination	85	9.9
	Psychological counseling	26	3
	Social support services	49	5.7
	Legal services	7	0.8
	Family services	20	2.3
	No answer	26	3
	None of these	490	57.2

## Data Availability

The data that support the findings of this study are available from the corresponding author upon reasonable request.

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
