# Peer review of "Street Children in Iran: What Are Their Living and Working Conditions? Findings from a Survey in Six Major Cities"

_ijerph, 2023, doi:10.3390/ijerph20075271_

Round 1

Reviewer 1 Report (Previous Reviewer 2)

The Authors in the study addressed the suggestions and comments identified in the review. The article requires editorial changes.

Author Response

Thank you for giving me the opportunity to submit a revised draft of our manuscript titled "Street children in Iran: What are their living and working conditions? Findings from a survey in six major cities?" to IJERPH. We appreciate the time and effort that you and the reviewers have dedicated to providing your valuable feedback on our manuscript. We are grateful to the reviewers for their insightful comments on our paper. We have been able to incorporate changes to reflect most of the suggestions provided by the reviewers. We have corrected them on the manuscript and responded all comments hear.

Reviewer 2 Report (Previous Reviewer 3)

The purpose of this study was to describe the street children population in several cities in Iran. The authors also sought to understand the difficulties and hazards that they were often exposed to. This paper has improved very significantly since I first reviewed it and it is now well-written with proper methodological detail to warrant publication. Thanks to the authors for responding so well to past review feedback.

Line 101: The sentence ends with “because” without outlying the reasons for criticism.

Line 114: Remove “of working” between problems and of

Line 112-134: Fix the formatting

Line 134: Replace the name of the first author with “elsewhere” and still reference the paper

Line 154-156: These aren’t really “outcomes” since you don’t model them with the other variables but rather hazards or health concerns.

Table 2: “Weighing people on scales without payment” – What does this mean? Can you provide some context to this in the text or as a footnote to the table?

Table 4: Please define DIC and Police 110

For all of the tables, it would be best if the numbers were right and justified for easier reading.

Line 245: Rather than “lack of educational cost” I assume you mean to say something like lack of funds to pay for education.

Line 276: missing a “to” in this sentence.

Line 269-279: The main point of this paragraph is to discuss that the metric of malnutrition/hunger in this study is not exactly the same as that used at the national level and thus hard to compare. Please clean the paragraph up to make that statement clearer and more direct.

Line 329: This statement is incorrect. You found a higher proportion of Afghan children than Iranian children in the population studied

A funding source is mentioned in the acknowledgments. It should be moved to funding. 

Author Response

Thank you for giving me the opportunity to submit a revised draft of our manuscript titled "Street children in Iran: What are their living and working conditions? Findings from a survey in six major cities?" to IJERPH. We appreciate the time and effort that you and the reviewers have dedicated to providing your valuable feedback on our manuscript. We are grateful to the reviewers for their insightful comments on our paper. We have been able to incorporate changes to reflect most of the suggestions provided by the reviewers. We have corrected them on the manuscript and responded all comments hear.

Line 101: The sentence ends with “because” without outlying the reasons for criticism.

Answer: Thank you for your comment, we have corrected it in the manuscript:

" But, these policies have been criticized because these did not have a significant outcome on their education and secondary needs"

Line 114: Remove “of working” between problems and of

Answer: Thank you for your comment, we have corrected it in the manuscript.

Line 112-134: Fix the formatting

Answer: Thank you for your comment, we have corrected it in the manuscript.

Line 134: Replace the name of the first author with “elsewhere” and still reference the paper

Answer: Thank you for your comment, we have corrected it in the manuscript.

Line 154-156: These aren’t really “outcomes” since you don’t model them with the other variables but rather hazards or health concerns.

Answer: Thank you for your comment, we have Replace "outcomes" with “health concerns”.

Table 2: “Weighing people on scales without payment” – What does this mean? Can you provide some context to this in the text or as a footnote to the table?

Answer: Thank you for your comment, actually it is "Weigh people on scales with payment". It is a kind of street job in which somebody provides a weighting scale and sits on the walkway and weighs people and earns a little money. I have added it as a footnote.

Table 4: Please define DIC and Police 110

Answer: Thank you for your comment, We have defined them as a footnote to the table:  Transition Center (DIC): To control or reduce high-risk behaviors, the center provides harm reduction services (healthy injection and healthy sexual behavior training, needle delivery) to people with substance abuse problems and hard-to-reach addicts who do not visit treatment centers.

Police 110 : Police 110 is a dispatcher center of Law Enforcement Command of Islamic Republic of Iran which sends emergency responder units

For all of the tables, it would be best if the numbers were right and justified for easier reading.

Answer: Thank you for your comment, It has done.

Line 245: Rather than “lack of educational cost” I assume you mean to say something like lack of funds to pay for education.

Answer: Thank you for your comment, we have Replace "lack of educational cost” with " lack of funds to pay for education".

Line 276: missing a “to” in this sentence.

Answer: Thank you for your comment, we have corrected it in the manuscript.

Line 269-279: The main point of this paragraph is to discuss that the metric of malnutrition/hunger in this study is not exactly the same as that used at the national level and thus hard to compare. Please clean the paragraph up to make that statement clearer and more direct.

Answer: Thank you for your comment, we have cleaned this paragraph up the manuscript.

Line 329: This statement is incorrect. You found a higher proportion of Afghan children than Iranian children in the population studied

Answer: Thank you for your comment. we have corrected it in the manuscript. Since, policymakers in Iran believe and insist that the most of children are immigrants and they aren’t Iranian, we found this argument is incorrect and their population approximately are equal.

A funding source is mentioned in the acknowledgments. It should be moved to funding. 

Answer: Thank you for your comment, It has done.

Reviewer 3 Report (New Reviewer)

The authors take up the extremely important topic of street children. However, the manuscript has some shortcomings that need to be refined. Introduction. At this point, there is no information about what is novelty about the work or what gap it fills.
2. method
Sampling and design
At this point, the respondents are described too generally. There is no information on how children of what age were selected for the study. Were they all doing the same job or something different? There is no information on the health condition of the children or how many hours a day they worked.

The sentence: "Data was gathered using a structured questionnaire based on the prior instrument in our previous study" is not clear. What you mean by prior instrument and previous study?

This point also lacks information on how the survey was constructed. In line 230 authors write that: "According to the ILO definition..." no abbreviation in the text. I recommend completing this item. The strong point of the work is a very interesting topic and well-conducted research. The weak point, however, is the rather chaotic presentation of the literature review. I recommend creating a separate Literature Review section, in which the authors will highlight what research on the topic has been carried out so far and how the literature relates to the topic. The authors in the work referred to only a few studies from 2022, so I also recommend supplementing this element.

Author Response

Thank you for giving me the opportunity to submit a revised draft of our manuscript titled "Street children in Iran: What are their living and working conditions? Findings from a survey in six major cities?" to IJERPH. We appreciate the time and effort that you and the reviewers have dedicated to providing your valuable feedback on our manuscript. We are grateful to the reviewers for their insightful comments on our paper. We have been able to incorporate changes to reflect most of the suggestions provided by the reviewers. We have corrected them on the manuscript and responded all comments hear.

  1. Introduction. At this point, there is no information about what is novelty about the work or what gap it fills.

Answer: Thank you for your comment, we have specified the research gap in the introduction.

  1. method
    Sampling and design
    At this point, the respondents are described too generally. There is no information on how children of what age were selected for the study. Were they all doing the same job or something different? There is no information on the health condition of the children or how many hours a day they worked.

Answer: Thank you for your comment.  We have explained the selection method on the manuscript:

Based on eligibility criteria, the participants included 10- to 18-year-old children who worked (any work) or had lived for a few hours a day for at least a month in the streets.

The information on the health condition of the children or how many hours a day they worked is in the result section.

The sentence: "Data was gathered using a structured questionnaire based on the prior instrument in our previous study" is not clear. What you mean by prior instrument and previous study?

Answer: Thank you for your comment.  Data were gathered using a structured questionnaire based on the prior instrument used in a previous study by the authors (53). The questionnaire included modules on socio-demographic characteristics, characteristics of work and problems of working on the street, and responses or intervention programs provided by governmental and non-governmental institutions. A total of 856 street children were interviewed. The response rate was 95.1%.

This point also lacks information on how the survey was constructed.

Answer: Thank you for your comment.  We selected a random sample of 464 venue-daytimes from the available venues, including corners of streets, parks, metro gates, bus stations, shopping malls, and shopping centers frequented by street children. then a brief interview was conducted with children who were systematically approached to gather data to verify living on the street, duration of being on the street, and demographic background.

 In line 230 authors write that: "According to the ILO definition..." no abbreviation in the text. I recommend completing this item.

Answer: Thank you for your comment, It has done.

The strong point of the work is a very interesting topic and well-conducted research. The weak point, however, is the rather chaotic presentation of the literature review. I recommend creating a separate Literature Review section, in which the authors will highlight what research on the topic has been carried out so far and how the literature relates to the topic. The authors in the work referred to only a few studies from 2022, so I also recommend supplementing this element.

Answer: Thank you so much, we have added new references in the introduction.

Round 2

Reviewer 3 Report (New Reviewer)

The authors did not address my recommendations, primarily regarding the literature review aspect. In its current form, I cannot agree to the publication of the manuscript, as it requires solid refinement.

Author Response

The authors did not address my recommendations, primarily regarding the literature review aspect. In its current form, I cannot agree to the publication of the manuscript, as it requires solid refinement.

Answer: Thank you for your comment, we have tried to address the research gap in the introduction more clearly in lines 123-141.

This manuscript is a resubmission of an earlier submission. The following is a list of the peer review reports and author responses from that submission.

Round 1

Reviewer 1 Report

The introduction does not focus sharply on the contribution of this paper. It starts with globalised generalisations, using terms like ‘child labour’ and ‘street children’ that carry a variety of meanings in different contexts. These general comments are neither informative nor particularly interesting in relation the data presented in this paper.

This paper reports on a general survey of children who spend some time on the streets of a number of Iranian cities, in order to suggest the kinds and scale of intervention that is appropriate. While the descriptive statistics are presented clearly enough, this function of the paper should be stated at the beginning to attract the attention of interested readers. The descriptive statistics are likely to be of interest for local practitioners and policy makers; but it is not clear that it has interest for an international readership, and it says little on the outcomes of work in children’s health, which is the focus of the special issue for which the paper was submitted.

The paper appears to argue from a general assumption that it is problematic to find children on the streets of a city and intervention is necessary. The paper needs to clarify what are the problems that need intervention, what kinds of intervention might help, and how widespread the various specific problems are. are.

The authors notice difficulty in defining the term ‘street children’. The research includes children who had ‘lived’ on the streets for ‘a few hours a day’. It is not clear what is the problem of children spending time in public places like streets for a few hours a day. The sample appears to include some enterprising young people who earn a little extra money on the streets – which is not a problem until people try to stop them; and on the other hand, it includes children forced to work long hours to the detriment of their schooling and even sometimes health – which clearly is a problem. So the conclusion that ‘street children require immediate attention’ needs to be refined: it is not evident from the survey data that all children caught in the survey need such attention.  The paper needs more discussion about how to focus on the problematic cases.

In fact, the vast majority of the children in the sample work for some kind of income. It would be helpful if the paper could engage some of the literature on children’s work and what constitutes harmful work. The ILO perspective is easy to find and summarise; some alternative perspectives can be found by consulting childrenandwork.net. The term ‘child labour’ carries different meanings in different contexts and care is needed when it is used. The conclusion of the paper speaks of ‘interventions to provide less harmful work environments and more monitoring for child laborers… who work on the streets’. While this appears a logical conclusion from findings that so many of the children are concerned with supporting their families, it needs to be discussed both as deriving from specific data produced by the survey, and in relation to the ILO policy of banning or restricting economic activity on the basis of age rather than on harm or benefits.

The discussion could say more about different categories of children met in the survey, and the different interventions that might be appropriate to these categories. For this kind of discussion, some cross tabulations might be useful. Is there a connection, for example, between parental educational background and the kind of work situations children are in? Or their educational level? What difference does living with parents or kin make to their situation? The paper reports cross tabulations to investigate differences between Afghan and Iranian children, but the significance for intervention of what differences emerged were not discussed.

Author Response

The introduction does not focus sharply on the contribution of this paper. It starts with globalised generalisations, using terms like ‘child labour’ and ‘street children’ that carry a variety of meanings in different contexts. These general comments are neither informative nor particularly interesting in relation the data presented in this paper.

Answer: Thanks a lot. We have revised the introduction of manuscript as you mentioned and highlight in the introduction. “Child labor” term eliminated and, we concentrated on “Child Street”. On the other hand, in previous version we focused sharply on child labor situation in Iran and editors asked us revised it, while, we tried to address in the new version. This paper present the street children's characteristics, the reasons that push them to the streets, their working conditions, and the challenges they might face in the streets in different areas of Iran. Therefore we gathered the relevance studies as background.

This paper reports on a general survey of children who spend some time on the streets of a number of Iranian cities, in order to suggest the kinds and scale of intervention that is appropriate. While the descriptive statistics are presented clearly enough, this function of the paper should be stated at the beginning to attract the attention of interested readers. The descriptive statistics are likely to be of interest for local practitioners and policy makers; but it is not clear that it has interest for an international readership, and it says little on the outcomes of work in children’s health, which is the focus of the special issue for which the paper was submitted.

Answer: This study helps to develop a dialog between academic researches and policy makers. However, the current phase of globalisation means child street problem may be understood differently by different governments, and there may be a requirement for some actions to be undertaken at the global level. While this paper reports on a general survey of children who spend some time on the streets of a number of Iranian cities, it has interest for an international readership because of: gaining access to more diverse facilities and participants in research and abling to undertake activities in another country that would not be permitted in this country, due to legal or ethical constraints.

The paper appears to argue from a general assumption that it is problematic to find children on the streets of a city and intervention is necessary. The paper needs to clarify what are the problems that need intervention, what kinds of intervention might help, and how widespread the various specific problems are.

Answer: Thanks a lot. We have revised the discussion of manuscript as you mentioned and highlight in the discussion.

The authors notice difficulty in defining the term ‘street children’. The research includes children who had ‘lived’ on the streets for ‘a few hours a day’. It is not clear what is the problem of children spending time in public places like streets for a few hours a day. The sample appears to include some enterprising young people who earn a little extra money on the streets – which is not a problem until people try to stop them; and on the other hand, it includes children forced to work long hours to the detriment of their schooling and even sometimes health – which clearly is a problem. So the conclusion that ‘street children require immediate attention’ needs to be refined: it is not evident from the survey data that all children caught in the survey need such attention.  The paper needs more discussion about how to focus on the problematic cases.

Answer: Thanks a lot. We have revised the introduction and discussion of manuscript as you mentioned and highlight in the introduction and discussion.

In fact, the vast majority of the children in the sample work for some kind of income. It would be helpful if the paper could engage some of the literature on children’s work and what constitutes harmful work. The ILO perspective is easy to find and summarise; some alternative perspectives can be found by consulting childrenandwork.net. The term ‘child labour’ carries different meanings in different contexts and care is needed when it is used. The conclusion of the paper speaks of ‘interventions to provide less harmful work environments and more monitoring for child laborers… who work on the streets’. While this appears a logical conclusion from findings that so many of the children are concerned with supporting their families, it needs to be discussed both as deriving from specific data produced by the survey, and in relation to the ILO policy of banning or restricting economic activity on the basis of age rather than on harm or benefits.

The discussion could say more about different categories of children met in the survey, and the different interventions that might be appropriate to these categories. For this kind of discussion, some cross tabulations might be useful. Is there a connection, for example, between parental educational background and the kind of work situations children are in? Or their educational level? What difference does living with parents or kin make to their situation? The paper reports cross tabulations to investigate differences between Afghan and Iranian children, but the significance for intervention of what differences emerged were not discussed.

Answer: Thanks a lot. We have revised discussion of manuscript as you mentioned and highlight in the discussion.

Reviewer 2 Report

The study deals with an important issue, which is the difficult situation of street children.

My insights to be developed:

- line 1 - the Authors should be specify the type of article

- lines 12, 262 - no period at the end of the sentence

- line 16 space between 2 and groups

- line 21 - space between 2 and groups

- line 22 - editing error

- line 25 - error in writing "

- lines 37, 40, 43, 54 and until the end of the work - no space between the words and the reference in the literature

- line 37 - it would be worth defining what dangerous work is, giving examples of it

1. Introduction - Authors should conduct a broader literature review. They should analyze the occurrence of similar problems in other countries, as well as propose solutions for actions that have an impact on reducing the issues discussed in the article. It is also worth pointing out what actions could be effective and improve the situation of children in Iran.

2. Method

- line 115 - Time-Location Sampling method - requires an explanation, the characteristics of the method

- line 127 - an interval between 12 and months

- variables - many editing errors, it would be worth presenting the records in a different form

- line 156 - error in writing []

- Analysis - the Authors should explain why they used the Chi square test

- line - 194 should be %

- table 1 – “out” should be capitalized

- table 2 - spaces between digits and parentheses

- table 2 - yrs. should be years

- table 2 - double space between Father's job;

- line 220 - no period after the “families”;

- 4. Discussion - this chapter requires the definition of directions for actions to improve the living conditions of children, references to proposed activities in this area, discussed in other studies and determining the directions of further research allowing for a deeper analysis of the topic.

- 5. Conclusions - this chapter requires development and reference to the results of work

References - editing errors - double numbering. Entries in the references are not in accordance with the requirements of the journal.

There are many editing errors at work. Authors should work on text editing and literature, in accordance with IJERPH guidelines.

I hope that my insights will be useful for the Authors of the study.

Author Response

The study deals with an important issue, which is the difficult situation of street children.

My insights to be developed:

- line 1 - the Authors should be specify the type of article

- lines 12, 262 - no period at the end of the sentence

- line 16 space between 2 and groups

- line 21 - space between 2 and groups

- line 22 - editing error

- line 25 - error in writing "

- lines 37, 40, 43, 54 and until the end of the work - no space between the words and the reference in the literature

- line 37 - it would be worth defining what dangerous work is, giving examples of it

Answer: Thanks a lot. We have revised manuscript as you mentioned and highlight.

  1. Introduction - Authors should conduct a broader literature review. They should analyze the occurrence of similar problems in other countries, as well as propose solutions for actions that have an impact on reducing the issues discussed in the article. It is also worth pointing out what actions could be effective and improve the situation of children in Iran.

Answer: Thanks a lot. We have revised the introduction of manuscript as you mentioned and highlight in the introduction.

  1. Method

- line 115 - Time-Location Sampling method - requires an explanation, the characteristics of the method

- line 127 - an interval between 12 and months

- variables - many editing errors, it would be worth presenting the records in a different form

- line 156 - error in writing []

- Analysis - the Authors should explain why they used the Chi square test

- line - 194 should be %

- table 1 – “out” should be capitalized

- table 2 - spaces between digits and parentheses

- table 2 - yrs. should be years

- table 2 - double space between Father's job;

- line 220 - no period after the “families”;

Answer: Thanks a lot. We have revised the method of manuscript as you mentioned and highlight in the method.

- 4. Discussion - this chapter requires the definition of directions for actions to improve the living conditions of children, references to proposed activities in this area, discussed in other studies and determining the directions of further research allowing for a deeper analysis of the topic.

Answer: Thanks a lot. We have revised the discussion of manuscript as you mentioned and highlight in the discussion.

- 5. Conclusions - this chapter requires development and reference to the results of work

Answer: Thanks a lot. We have revised the conclusion of manuscript as you mentioned and highlight in the conclusion.

References - editing errors - double numbering. Entries in the references are not in accordance with the requirements of the journal.

There are many editing errors at work. Authors should work on text editing and literature, in accordance with IJERPH guidelines.

Answer: Thanks a lot. We have revised all editing errors in the manuscript and highlight.

I hope that my insights will be useful for the Authors of the study.

Reviewer 3 Report

Summary of comments:

The main goal of this paper is to provide better estimates, information, characteristics, and statistics to describe the street children population in Iran. There are no hypotheses stated in the introduction as it is presented in the intro as descriptive in nature. The paper is written on an important and understudied topic.

Although it is presented as a descriptive paper, the methods and much of the results discuss the differences between Afghan and Iranian street children in Iran, however, the single paragraph in the introduction about migrants and refugees fails to fully set up why assessing the characteristics by nationality is important. The paragraph only references discrepancies in the proportion of Iranian children by province/city. None of the results are presented by province or city in this paper so it is hard to connect this information more directly to the existing literature. The results, particularly the tables, could be better presented. For example, if the main premise of the paper is to understand how Iranian and non-Iranian street children differ in Iran to help inform targeted interventions and change the political dialogue about street children then the results need to be presented better. I suggest all variables be presented by nationality so that chi-squared results can be directly reported in the Tables. Further, the methods need to be more detailed. It is unclear from the written methods exactly how the questions are asked and which questions allow for multi-answers. Often the percentages reported in the text do not accurately represent the prevalence of a reported selection but rather the proportion of all possible responses across all individuals which is not informative. Due to the sampling design, daytime-TLS, there are significant and potential biases/limitations that need to be mentioned and discussed in the discussion, but there is no limitation section. The discussion should be expanded to discuss rather than just state the differences in this research from the literature. What might be causing the differences seen. Finally, the authors briefly bring up the misconceptions of street children and the political discourse around them, but then they just jump to interventions being generally important. I think a much more powerful and potentially impactful conversation could happen in the discussion if specific interventions and policy implications were called out and discussed in the context of the new information and understanding gathered by this research.

Of note, there are many punctuation, capitalization, and editorial issues throughout the paper and it proofreading for English language throughout.

Specific Comments:

Title: The title on the document does not match the title in the review submission portal. Since the sample includes both Afghan and Iranian children, I suggest you go with the one in the submission portal.

Abstract:

Line 16: change 2 to “two” and add space.

Line 17: Change to 90% of the participants were boys.

Line 17 and 20: remove “according to the” and instead say findings showed… or Chi-square analysis showed.

Line 21: I would suggest just referring to Afghan and Iranian children rather than “2groups”. Remove “2groups” throughout. Remove “of children” from the end of the sentence as well.

Line 24: “from” instead of “of”

Line 25-26: “were not involved” instead of “did not involve”.

Line 27: “reported no involvement in” instead of “reported none involved”

Introduction:

Line 41-42: consider revising the last two sentences to one that says “but it is likely that their increase parallels population growth, internal migration, and urbanization and has been increasing worldwide, particularly in developing countries.”

Line 56: Instead of “faces them” use “subjects them to”. Change this usage throughout the text.

Line 57-59: remove extra spaces and add an “and” in front of “also experience lower..”

Line 72: Instead of using these throughout when referring to items in a previous sentence. Several editorial errors in this paragraph.

Line 97-99: This sentence is unclear and adds nothing to the introduction

Method:

What is ta rapid assessment and response method? If this is a standardized method, please site the reference for design.

Lines 124-129: There are several editorial and typo issues in this paragraph.

Variables section: lines 133-137 should not be here. Add whatever eligibility criteria is listed here to the above paragraph that talks about eligibility. Start with the first sentence and end it at “children” then continue with “six health outcomes…”

Line 141: What are “agents”? How were they defined for the participants?

Line 142: Why is “The Hunger Experience” capitalized? Is this the name of an official survey tool that defines experiencing hunger or just a question… “have you in the last 12 months experienced hunger?” Please clarify the exact design of these questions further. Also move the sentence about “all behavioral questions referred to the 12 months prior” to this section as it is a more relevant place.

Lines 143-158: Please clear up the editorial issues in this paragraph. Was nationality only asked as either Afghan or Iranian? Was there no potential to sample anyone of a different nationality? If it was an inclusion/exclusion criteria please list it as such. What is considered a drug, how is this interpreted or was it explained to the participants? Was mother’s education not mentioned (it is presented in Table 1)? Consider splitting this paragraph into a few sentences. What is “conventional form” of work? Can you please present what is meant for arrested in the last 3 months (categories by agent) and the street jobs listed? Even if some of the details about the survey and questions are relegated to the Appendix/supplemental documents, more details must be included.

Analysis Section: It is unclear why comparisons between Afghan and Iranian children was done? Was this an a priori hypothesis difference? If so, the introduction needs to expand upon the importance of nationality differences among street children and what that could mean. Also, just use “Afghan and Iranian” rather than “2groups (Afghan and Iranian)”, as it’s redundant and takes up unnecessary space without providing any clarity. Why were the chi-squared differences between Afghan and Iranian street children only tested for a very small subset of the questions? Again was this a priori hypotheses? No a priori hypotheses were outlined in the introduction, instead this paper was framed as an exploratory and descriptive assessment of street children. The analysis should reflect that and thus describe and assess all variables collected.

Ethics Section: Please revise this section, it is very unclear. Were informed consents actually collected or just verbal okays? What University is the Ethics Board housed under?

Results:

Line 186: This is not a significant statistic that is reported. Please revise.

Line 186-187: This sentence does not make sense as written. Since this is a cross-sectional study the wording needs to read as such. Consider revising to “The 42.1% of children with addicted fathers were less likely to report that they were currently enrolled in school (Chi square…”

Working conditions section: Were participants asked to respond with all of the types of street work they did in the past 12 months? If so, please indicate that in this section and make it more clear in the methods. The percentages reported in this section are sometimes based on the count of persons surveyed and sometimes based on all of the answers collected for a multi-selection question. This approach for presenting the results and the lack of clarity about when it differs is very misleading. See my notes about Table 3.

Health outcomes faced by street children section: The N here is 810? Were participants only allowed to identify one health outcome that they may have suffered in x (what is the timeframe) amount of time?

Interventions and services section: Please make clear which questions allow for multi-select and which do not. Consider presenting the combinations that were reported to make all tables and percentages reported person-level characteristics.

Discussion:

Lines 266-268: Which provinces? Are you able to show data about provinces? Maybe even just how many children were recruited from each?

Line 284: “end” should be “and”

Line 285-286: text past the word “Iran” is unclear as to what it is trying to convey.

Please add a section on limitations. The Time location sampling for example has limitations in that locations were only selected for daytime recruitment thus potentially biasing the population sampled. Please discuss this and draw from appropriate literature.

Conclusions:

Consider revising the first sentence to read “ Contrary to dominant labels and political discourse, street children in Iran are mainly concerned with supporting their families and are not predominantly of immigrant families.”

I think the most important conclusions of this study are the statistics that are counter to the general knowledge or political rhetoric. The differences seen are then important considerations for promising or future interventions.

References: The formatting needs fixing as the numbers for each reference in the list are duplicated.

All Tables: For all tables consider being a bit more descriptive in the variable. Since there is so much space you could just include the question asked, the timeframe assessed, and the N that responded in each situation. Use footnotes to describe some of the items in the table that are not well-defined. For example “Transit Centers (DIC)” in Table 5… what does that mean? Consider including the survey questionnaire in the Appendix/Supplemental Materials. Also, please verify that all variables reported in the tables are described in the methods and vis versa. 

Table 1: The N next to the variables in the table does not match that reported in the text, in the Table title, or in the count sums. Please correct accordingly. If your primary hypothesis is that there will be differences by nationality. I would consider presenting Table 1 as a stratified presentation. Have side-by-side columns of nationality and the count (%) of each demographic/behavioral/etc. variable under it (basically like you already have in Table 2, just don’t limit it to SES related variables). Effectively a stack of two-by-two tables that you can also show the chi-squared results across nationality for each variable in. It would be a very succinct way to display all of the descriptive data in one Table rather than two. Particularly since it is unclear why the specific variables were chosen for inclusion in Table 2 and subsequent statistical testing.

Table 2: Consider combining with Table 1 to show all information together and concisely.

Table 3: Consider presenting these variables by nationality as well, if this is a primary focus of the paper. It seems that participants were able to “select all that apply” for the “Most important reasons for working”, if that is correct please make that clear in the methods and consider reporting the frequency of answer combinations in the table instead. Like that X number of participants report family support and personal needs as most important. Please apply the same presentation methods for other mulit-selection questions.

Table 4: The line about “did not have a place to sleep” is missing from the table. Please verify the N in the table title.

Table 5: Similar comments to the other 4 tables. 

Author Response

The main goal of this paper is to provide better estimates, information, characteristics, and statistics to describe the street children population in Iran. There are no hypotheses stated in the introduction as it is presented in the intro as descriptive in nature. The paper is written on an important and understudied topic.

Although it is presented as a descriptive paper, the methods and much of the results discuss the differences between Afghan and Iranian street children in Iran, however, the single paragraph in the introduction about migrants and refugees fails to fully set up why assessing the characteristics by nationality is important.

Answer: Thanks a lot. We have revised the introduction of manuscript as you mentioned and highlight in the introduction.

The paragraph only references discrepancies in the proportion of Iranian children by province/city. None of the results are presented by province or city in this paper so it is hard to connect this information more directly to the existing literature. The results, particularly the tables, could be better presented. For example, if the main premise of the paper is to understand how Iranian and non-Iranian street children differ in Iran to help inform targeted interventions and change the political dialogue about street children then the results need to be presented better. I suggest all variables be presented by nationality so that chi-squared results can be directly reported in the Tables. Further, the methods need to be more detailed. It is unclear from the written methods exactly how the questions are asked and which questions allow for multi-answers. Often the percentages reported in the text do not accurately represent the prevalence of a reported selection but rather the proportion of all possible responses across all individuals which is not informative. Due to the sampling design, daytime-TLS, there are significant and potential biases/limitations that need to be mentioned and discussed in the discussion, but there is no limitation section. The discussion should be expanded to discuss rather than just state the differences in this research from the literature. What might be causing the differences seen. Finally, the authors briefly bring up the misconceptions of street children and the political discourse around them, but then they just jump to interventions being generally important. I think a much more powerful and potentially impactful conversation could happen in the discussion if specific interventions and policy implications were called out and discussed in the context of the new information and understanding gathered by this research.

Answer: Thanks a lot. We have revised the introduction, method, result and discussion of the manuscript as you mentioned and highlight in the manuscript. On the other hand, the purpose of this study is not investigation the difference between Iranian and Afghan children, and we did not sampling based on the population of Afghan and Iranian children. After discussion with the authors and the methodologist of this project, we have realized it is not possible to analyze this data between Iranian and Afghan children.

Of note, there are many punctuation, capitalization, and editorial issues throughout the paper and it proofreading for English language throughout.

 Answer: Thanks a lot. We have revised all editing errors in the manuscript and highlight.

Specific Comments:

Title: The title on the document does not match the title in the review submission portal. Since the sample includes both Afghan and Iranian children, I suggest you go with the one in the submission portal.

Abstract:

Line 16: change 2 to “two” and add space.

Line 17: Change to 90% of the participants were boys.

Line 17 and 20: remove “according to the” and instead say findings showed… or Chi-square analysis showed.

Line 21: I would suggest just referring to Afghan and Iranian children rather than “2groups”. Remove “2groups” throughout. Remove “of children” from the end of the sentence as well.

Line 24: “from” instead of “of”

Line 25-26: “were not involved” instead of “did not involve”.

Line 27: “reported no involvement in” instead of “reported none involved”

Answer: Thanks a lot. We have revised all editing errors in the manuscript and highlight.

Introduction:

Line 41-42: consider revising the last two sentences to one that says “but it is likely that their increase parallels population growth, internal migration, and urbanization and has been increasing worldwide, particularly in developing countries.”

Line 56: Instead of “faces them” use “subjects them to”. Change this usage throughout the text.

Line 57-59: remove extra spaces and add an “and” in front of “also experience lower..”

Line 72: Instead of using these throughout when referring to items in a previous sentence. Several editorial errors in this paragraph.

Line 97-99: This sentence is unclear and adds nothing to the introduction

Answer: Thanks a lot. We have revised all editing errors in the manuscript and highlight

Method:

What is ta rapid assessment and response method? If this is a standardized method, please site the reference for design.

Answer: Thanks a lot. We have revised the method of manuscript as you mentioned and highlight in the method.

Lines 124-129: There are several editorial and typo issues in this paragraph.

Variables section: lines 133-137 should not be here. Add whatever eligibility criteria is listed here to the above paragraph that talks about eligibility. Start with the first sentence and end it at “children” then continue with “six health outcomes…”

Answer: Thanks a lot. We have revised the introduction of manuscript as you mentioned and highlight in the introduction.

Line 141: What are “agents”? How were they defined for the participants?

Answer: Arresting by governmental agents is a common experience for children, nearly 45% of them at least had an experience of arresting by the municipality, police department, and State Welfare Organization when working in the streets

Line 142: Why is “The Hunger Experience” capitalized? Is this the name of an official survey tool that defines experiencing hunger or just a question… “have you in the last 12 months experienced hunger?” Please clarify the exact design of these questions further. Also move the sentence about “all behavioral questions referred to the 12 months prior” to this section as it is a more relevant place.

Answer: The hunger experience is just a question:  “have you in the last 12 months experienced hunger?”

Thanks a lot. We have revised it and moved the sentence about “all behavioral questions referred to the 12 months prior” to this section as you as you mentioned and highlight in the method.

Lines 143-158: Please clear up the editorial issues in this paragraph. Was nationality only asked as either Afghan or Iranian? Was there no potential to sample anyone of a different nationality? If it was an inclusion/exclusion criteria please list it as such. What is considered a drug, how is this interpreted or was it explained to the participants? Was mother’s education not mentioned (it is presented in Table 1)? Consider splitting this paragraph into a few sentences. What is “conventional form” of work? Can you please present what is meant for arrested in the last 3 months (categories by agent) and the street jobs listed? Even if some of the details about the survey and questions are relegated to the Appendix/supplemental documents, more details must be included.

Answer: Thanks a lot. The purpose of this study is not investigation the difference between Iranian and Afghan children, and we did not sampling based on the population of Afghan and Iranian children. After discussion with the authors and the methodologist of this project, we have realized it is not possible to analyze this data between Iranian and Afghan children and removed table 1.

Analysis Section: It is unclear why comparisons between Afghan and Iranian children was done? Was this an a priori hypothesis difference? If so, the introduction needs to expand upon the importance of nationality differences among street children and what that could mean. Also, just use “Afghan and Iranian” rather than “2groups (Afghan and Iranian)”, as it’s redundant and takes up unnecessary space without providing any clarity. Why were the chi-squared differences between Afghan and Iranian street children only tested for a very small subset of the questions? Again was this a priori hypotheses? No a priori hypotheses were outlined in the introduction, instead this paper was framed as an exploratory and descriptive assessment of street children. The analysis should reflect that and thus describe and assess all variables collected.

Answer: Thanks a lot. The purpose of this study is not investigation the difference between Iranian and Afghan children, and we did not sampling based on the population of Afghan and Iranian children. After discussion with the authors and the methodologist of this project, we have realized it is not possible to analyze this data between Iranian and Afghan children and removed table 1.

Ethics Section: Please revise this section, it is very unclear. Were informed consents actually collected or just verbal okays? What University is the Ethics Board housed under?

Answer: Thanks a lot. We have revised the method of manuscript as you mentioned and highlight in the method.

Results:

Line 186: This is not a significant statistic that is reported. Please revise.

Line 186-187: This sentence does not make sense as written. Since this is a cross-sectional study the wording needs to read as such. Consider revising to “The 42.1% of children with addicted fathers were less likely to report that they were currently enrolled in school (Chi square…”

Working conditions section: Were participants asked to respond with all of the types of street work they did in the past 12 months? If so, please indicate that in this section and make it more clear in the methods. The percentages reported in this section are sometimes based on the count of persons surveyed and sometimes based on all of the answers collected for a multi-selection question. This approach for presenting the results and the lack of clarity about when it differs is very misleading. See my notes about Table 3.

Answer: Thanks a lot. We have revised the results of manuscript as you mentioned and tried to make it more clear in the method and result.

Health outcomes faced by street children section: The N here is 810? Were participants only allowed to identify one health outcome that they may have suffered in x (what is the timeframe) amount of time?

Answer: Thanks a lot. We have revised the results and method of manuscript. The N here is 856. The children allowed to identify more than one health outcome that they may have suffered in the 12 months prior.

Interventions and services section: Please make clear which questions allow for multi-select and which do not. Consider presenting the combinations that were reported to make all tables and percentages reported person-level characteristics.

Answer: Thanks a lot. We have revised the results of the manuscript as you mentioned.

Discussion:

Lines 266-268: Which provinces? Are you able to show data about provinces? Maybe even just how many children were recruited from each?

Line 284: “end” should be “and”

Line 285-286: text past the word “Iran” is unclear as to what it is trying to convey.

Answer: Thanks a lot. We have revised all editing errors in the manuscript and highlight

Please add a section on limitations. The Time location sampling for example has limitations in that locations were only selected for daytime recruitment thus potentially biasing the population sampled. Please discuss this and draw from appropriate literature.

Answer: Thanks a lot. We have added a section on limitations in the manuscript as you mentioned.

Conclusions:

Consider revising the first sentence to read “ Contrary to dominant labels and political discourse, street children in Iran are mainly concerned with supporting their families and are not predominantly of immigrant families.”

I think the most important conclusions of this study are the statistics that are counter to the general knowledge or political rhetoric. The differences seen are then important considerations for promising or future interventions.

Answer: Thanks a lot. We have revised the conclusion of the manuscript as you mentioned.

References: The formatting needs fixing as the numbers for each reference in the list are duplicated.

Answer: Thanks a lot. We have revised the references of the manuscript as you mentioned.

All Tables: For all tables consider being a bit more descriptive in the variable. Since there is so much space you could just include the question asked, the timeframe assessed, and the N that responded in each situation. Use footnotes to describe some of the items in the table that are not well-defined. For example “Transit Centers (DIC)” in Table 5… what does that mean? Consider including the survey questionnaire in the Appendix/Supplemental Materials. Also, please verify that all variables reported in the tables are described in the methods and vis versa. 

Answer: Service Provider Units Injury reduction programs are offered according to the needs and specifications of the area and evaluations performed in various units, which are: Transit Center (DIC), Night Shelter Center (Shelter), Passage Center / Night Shelter (DIC / shelter), Team Outreach, Mobile Center, Injury Reduction Station. Transition Center (DIC): In order to control or reduce high-risk behaviors, the center provides harm reduction services (healthy injection and healthy sexual behavior training, needle delivery) to people with substance abuse problems and hard-to-reach addicts who do not visit treatment centers. Offers syringes, condoms, food, clothing and bathing). These centers are daily and provide services 4 to 6 hours a day. In 26 provinces, there are 96 men's transit centers and 12 women's transit centers in 9 provinces

Table 1: The N next to the variables in the table does not match that reported in the text, in the Table title, or in the count sums. Please correct accordingly. If your primary hypothesis is that there will be differences by nationality. I would consider presenting Table 1 as a stratified presentation. Have side-by-side columns of nationality and the count (%) of each demographic/behavioral/etc. variable under it (basically like you already have in Table 2, just don’t limit it to SES related variables). Effectively a stack of two-by-two tables that you can also show the chi-squared results across nationality for each variable in. It would be a very succinct way to display all of the descriptive data in one Table rather than two. Particularly since it is unclear why the specific variables were chosen for inclusion in Table 2 and subsequent statistical testing.

Answer: Thanks a lot. Similar answer to the method section about Iranian and Afghan children investigation.

Table 2: Consider combining with Table 1 to show all information together and concisely.

Answer: Thanks a lot. Similar answer to the method section about Iranian and Afghan children investigation

Table 3: Consider presenting these variables by nationality as well, if this is a primary focus of the paper. It seems that participants were able to “select all that apply” for the “Most important reasons for working”, if that is correct please make that clear in the methods and consider reporting the frequency of answer combinations in the table instead. Like that X number of participants report family support and personal needs as most important. Please apply the same presentation methods for other mulit-selection questions.

Answer: Thanks a lot. Similar answer to the method section about Iranian and Afghan children investigation. And, we have revised the tables of the manuscript as you mentioned.

Table 4: The line about “did not have a place to sleep” is missing from the table. Please verify the N in the table title.

 Answer: Thanks a lot. We have revised the table of the manuscript as you mentioned.

Table 5: Similar comments to the other 4 tables. 

Answer: Thanks a lot. We have revised the table of the manuscript as you mentioned.

Round 2

Reviewer 1 Report

Although the revised version responds to some my concerns, it fails to deal adequately with key issues surrounding children living or working in public places. There is still very little analysis of the relationships between different variables surveyed. The result is that the discussion is very generalized, and provides little to indicate specific problems of children in different situations.

The presentation gives the impression that the authors take for granted norms of better-resourced communities to determine what childhood should be, and assume childhoods that are different to be problematic. It makes sweeping generalisations of the “problem” of street children, failing to differentiate those children using the public places of the streets constructively from those whose situation on the streets are damaging their chances.

 For example, it offers a UN definition of street children including the phrase “… and who are inadequately protected or supervised by responsible adults”. We do not find a discussion of what kind of supervision or protection might be regarded as adequate in different kinds of activities, nor what kinds of supervision or protection are experienced by members of the sample.

Another example is the consideration of children’s work. The paper speaks of work that “deprives them (any person under 18) of their childhood, potential, and dignity, which harms their physical and/or mental development”. What does it mean to deprive someone of their childhood? Many in the sample have to work long hours, sometimes in hazardous conditions, depriving them of schooling. But the sample also includes children who spend only a few hours a day on the streets, a few working less than two hours a day: even if their work is classified by ILO as “child labor” on the basis of age and economic activity, there is no evidence offered that such work is in any way harmful and demanding intervention.

The presentation of health outcomes states that children “mostly have suffered …”, and yet each of the following health categories show only a minority of the sample suffering, and there is no indication how these percentages compare with other children who are not found on the streets. For example, some studies in other countries have shown that working children sometimes have better nutrition than children in their communities who do not work; so when you say 20.7% report “starving”, it is possible the figure in their communities would be higher for those not working on the streets, in which case being on the streets is not so much a problem as a way of mitigating a greater problem.

In relation to children’s reasons for working, other studies have found that some children want to work to acquire useful experience, or for enjoyable social reasons. Were there no such responses in this sample? Or perhaps such responses were not recorded as relevant to the study?

The paper mentions that current policies have been criticised, but does not discuss these policies nor the criticisms of them. It does not discuss which children need immediate attention and what different kinds of attention are appropriate for children in different situations. Overall, I have not changed my original assessment: this survey could provide a useful basis for discussion and brainstorming among local practitioners and policy makers along the lines of the final paragraph of the paper, but it has very limited interest for an international readership.

Author Response

  1. Please address round 2 comments, especially: What does it mean to deprive someone of their childhood? Many in the sample have to work long hours, sometimes in hazardous conditions, depriving them of schooling. But the sample also includes children who spend only a few hours a day on the streets, a few working less than two hours a day: even if their work is classified by ILO as “child labor” on the basis of age and economic activity, there is no evidence offered that such work is in any way harmful and demanding intervention.

Answer: Thank you for your question. Based on our previous study (https://www.ncbi.nlm.nih.gov/pmc/articles/pmid/30963510/), without hours work consideration, street children are vulnerable group and they are at risk of health problems.

However, in this study there are three inclusion criteria: 1) children age between 10 to 18 years, 2) spending hours on the street, 3) spending hours on the street at least for a month.

  1. Please try in the discussion to distinguish between the problematic cases that require intervention (and which types of interventions) compared to cases that are not harmful.

Answer: Thank you for your comment. We have considered this valuable suggestion in the discussion part.

  1. The presentation of health outcomes states that children “mostly have suffered …”, and yet each of the following health categories show only a minority of the sample suffering, and there is no indication how these percentages compare with other children who are not found on the streets. For example, some studies in other countries have shown that working children sometimes have better nutrition than children in their communities who do not work; so when you say 20.7% report “starving”, it is possible the figure in their communities would be higher for those not working on the streets, in which case being on the streets is not so much a problem as a way of mitigating a greater problem.

Answer: Thank you for your comment. We have improved this section in our manuscript.

  1. Please include this point and cite research on that compares street children and non-street children where relevant to your findings (e.g. drug use - https://journals.sagepub.com/doi/abs/10.1177/1010539510361515), perhaps here as well - https://www.jahonline.org/article/S1054-139X(13)00188-2/pdf

Answer: Thank you for your suggestion. We have added these valuable references in our manuscript.

Reviewer 3 Report

The paper is improved. I think that the conclusion section touches best on the topics that should be better discussed in the discussion section. I think that the refusal to directly (in form of a thesis) compare and contrast characteristics by nationality is underwhelming particularly since there is a considerable amount of the discussion that references it. The edits were a little hard to follow because they were not all track changed. It would be much nicer for reviewers in the future if the revisions include tracked changes for all things changed. I think that if the authors present the results by nationality (even without statistical testing) and expand upon that and the policy/societal implications in the discussion this paper would warrant publication. Finally, there are still many punctuation, editing, and grammatical errors throughout that need to be addressed before publication.

Abstract:

Line 14: Children should not be capitalized.

The first conclusion line is inconsistent with the previous line. Consider “little involvement” inside of “no involvement”

Introduction:

Thank you for making the changes suggested by both reviewers. It is much better

Line 53: There should not be a comma after “child neglect” as I assume you are trying to say “child neglect and abuse”.

Line 57: Please either list the African countries or summarize the Indonesia, Bangladesh, and Nepal studies regionally as well OR site the number of African countries referenced. This sentence reads very poorly as is and is not inclusive.

Line 61: Please choose a different word from “announce” – maybe observed.

Line 63-64: Please revise the last sentence. Currently it is combining things that are experienced with things that are done by the person. Consider “Futhermore, they experience sexual abuse and violence, and are more likely to participate in risky activities that may include exposure to STIs.”

Line 66-67: consider “is described in many publications” also consider “five action areas across three levels – global, local, and personal”

Line 76-77: Consider “with a population of more than 80 million,”

Line 82: Change “on the other side” to “however”

Line 83: Change this sentence to read: “; while a study across four provinces showed that nearly 70% of street children were between 6 and 14 years of age.”

Line 84-85: This sentence doesn’t make sense and has a lot of unnecessary information in it. Trim it to the point you are trying to convey.

Lines 86-88: It’s often more helpful to the reader to say what you want to say and then cite the stats you have found to support/or contradict what you want to say. Here’s an example: “There are mixed reports of the proportion of street children in Iran that are Iranian; Tehran in 2012 reported about 36% non-Iranian, a study across four provinces in 2015 reported a much lower prevalence of approximately 16%, while a study in Kermanshah City in 2017 reported 100% Iranian street children. These studies highlight that location may also play an important role in the demographics of street children.” Or generally more like this. The introduction is good, but could use some connection of the facts in between to help the reader remember it more like a story.

Lines 81-89: Fix spacing issues throughout

Lines 92-95: this is another spot where the start of the sentence doesn’t really tell us much but could be written more informatively. Such as “Differences in have been seen by nationality, a study showed that Iranian street children…, while”

Lines 96-104: This is a massive run-on sentence. Consider reframing to “There are many policies and measures that exist to protect street children in Iran, including…” However, since 1999 more work has been actively done and includes the establishment of centers (…) and schools, and the development of protocols for gathering, managing, and supporting them, etc…” Although there are many policies, it is worth noting that there is considerable criticism of them, their implementation and evaluation.”

Line 133: remove extra period.

Line 134: This response rate seems very high particularly given the population being studied. How was this calculated/determined?

Line 146: extra space and “potential” is erroneously capitalized.

Line 151: Typo is married

Line 152: Missing comma

Line 153: extra spaces

Lines 146-157: extra spaces, no spaces, typos, missing commas, things capitalized that should not be, tense problems all throughout. Please fix.

Lines 176-178: The percentages reported here is the text don’t make sense. The text states that 48% were out of school and of this group 24.8% never attended school, but then the next sentence says that most out of school children (94.7%) quit education in elementary and middle school period. These statements are not possible as 48% + 94.7% >100%. Please correct either the numbers or the words to reflect what is correct.

Lines 174-183: Why only present the statistical results for boy and girls enrolled in school and addiction? In the response to reviewers it was indicated that upon further conversations with the statistical consultant statistical testing was not advised.

Table 1: Marital status has no n next to it, please update, similarly for Father’s job status, Mother’s job status, Fathers’ educational status, Mothers educational status, Father’s drug use, Mother’s drug use, Sleeping place in recent month, Children’s Identity document

Table 4: What is the “110 police” mean?

Lines 233-238: Run-on sentence without a clear point. Similarly with the next sentence; it is not clear.

Line 244: “lack of educational costs” doesn’t make sense.

Line 253-255: Doesn’t make sense, particularly “avoidance of challenges”. Similarly Lines 259-261. Please clear these sentences up so they more directly make the point attempting to be made.

Discussion: A large section of the discussion is dedicated to the Iranian/Afghan differences but that was still no clear thesis of this in the introduction and the fact that none of the results are shown by nationality is also misleading to the purpose of the paper. If it is a main topic you want to discuss (and I think that it is an important one, even if you don’t perform statistical tests across groups) then the results need to be presented accordingly.

Conclusion: The content in the conclusion is good, but in the wrong place. The topics brought up in the conclusion section should be first mentioned in the discussion and expanded upon. Further, the limitations should be in the discussion and are more than just the two listed here. Also, recall bias does not always lead to underreporting as it can also lead to overreporting. Finally, this conclusion is far too long. Move these topics and their discussion to the discussion section and briefly summarize all the things in the conclusion.

Funding section: The State Welfare Org of Iran and the University of Social Welfare and Rehab should be listed here not in the acknowledgments. It’s unclear what the China Medical Board Foundation grant funding is and whether it covered the study or just the publication…

Author Response

  1. Now the tables by nationality have been removed, it looks odd that the Discussion has a large section on differences by nationality (lines 248-269).

Answer: Thank you for your comment. We have summarized this phase in our manuscript

  1. Please address Reviewer’s 3’s round 2 comments on this - we’d suggest you present the results by nationality as in the original paper, and expand on the policy/societal implications in the Discussion, such as specific interventions for Afghan compared to Iranian children.

Answer: Thank you for your suggestion. The authors didn’t agree to present the result by nationality, because of:

1) There is no significant difference between Iranian and Afghan children on the outcome variables (you can see the original paper). We just added this part in our manuscript to respond reviewer’s comments.

2) For presenting the result by nationality, the whole structure of this paper must be change.

3) Because of there is no specific intervention based on children's nationality, the difference between Iranian and Afghan street children is not our prepose in this manuscript.

  1. Please address the grammatical and language edits in round 2 comments.

Answer: Thank you for your comment. We edited English language and revised language mistakes.

  1. Sampling and design section – please address our initial comments: How was the survey instrument designed (e.g. based on prior studies or instruments)? How did you obtain a listing of the 464 venues from which you conducted random sampling to locate children? Please also include details about the qualitative phase (was this separate research or purely to identify study sites for the survey?)

Answer: Thank you for your comment. In this study we used semi-structured questioner that was used in our previous study. we have corrected this in the manuscript.

The purpose of the qualitative phase was to create a list of all potential venues where street children could be found and identify the days and time periods when the maximum number of street children are present. Key informants included diverse persons with knowledge of street children at the city level (e.g., municipal social welfare personnel, public and non-governmental service providers, academicians) and the street or venue level (e.g., street children themselves). In practice, the group discussions with street children corroborated venues named by other key informants and added venues not previously identified in each city. Venues solicited included corners of streets, parks, metro gates, bus stations, shopping malls or centers frequented by street children.

Key informants and group discussants identified 370 venues (range 23 to 113 per city) where street children were purportedly present . Interviews with street children in the field identified an additional 94 venues (range 4 to 32 across cities). Of the 464 total venues, our team visited 226 (48.7%) at the randomly selected venue-day-times for the 1-h counting periods , and 200 venues (43.1%) for the 4-h periods.

for more information: https://www.ncbi.nlm.nih.gov/pmc/articles/pmid/30963510/

  1. Variables section – there is no need to state each variable response option in brackets (e.g. ‘Fathers job status (employed/retired/unemployed)) – please remove response options.

Answer: Thank you for your comment. Response options are removed.

  1. Ethics section – please specify what gift was offered to participants.

Street children attending group discussions received food and refreshments. Street children interviewed at venues received 50,000 Rials (US$1.35) for their time (in 2017).

  1. Please also address our initial comments: Specify what the informed consent process involved with children, and any special measures to ensure they did not feel coerced to participate. Please include how any safety issues were assessed (e.g. do children have ‘gatekeepers’ that restrict access to them? Did you approach gatekeepers to request children’s participation?)

Answer: Thank you for your important consideration. We explained the study to the children verbally then we obtained a verbal informed consent. Due to we didn’t access to gatekeepers, we didn’t obtain their consent.

In our study the protection of children included a range of issues. We have used UNICEF Ethical reporting guidelines principles and Guidelines for interviewing children (https://www.unicef.org/eca/media/ethical-guidelines). we have consideraed these principles:

Do no harm to any child; avoid questions, attitudes or comments that are judgmental, insensitive to cultural values, that place a child in danger or expose a child to humiliation, or that reactivate a child's pain and grief from traumatic events.

Do not discriminate in choosing children to interview because of sex, race, age, religion, status, educational background or physical abilities.

Explain the purpose of the interview and its intended use.

Furthermore, we have Trained the researchers:  This included the ability to assess the situation and any risks to the participant and refer them to the organizations for necessary interventions.

  1. Limitations should come at the end of the Discussion, before the Conclusion, and be more substantial than 2 lines. What other limitations are there of these data and the analysis? This section should be at least one full paragraph. See an example here (last paragraph before the Conclusion) – https://www.mdpi.com/1660-4601/19/12/7405. For example, there was low prevalence of sexual abuse disclosed. Often, survey questions are poorly phrased and interviewers not sufficiently trained to elicit accurate information on sexual abuse, and perhaps you could reference methods papers on this e.g. https://trialsjournal.biomedcentral.com/articles/10.1186/s13063-015-1004-7

Answer: Thank you for your comment, we have corrected it in the manuscript.